# Our perception may not be reality: A longitudinal study of the relationship between perceived and actual change in smoking behavior

**Astrid Juhl Andersen** [ID]*, **Solène Wallez** [ID], **Maria Melchior** [ID], **Murielle Mary-Krause** [ID]

INSERM, Institut Pierre Louis d'Épidémiologie et de Santé Publique, IPLESP, Equipe de Recherche en Epidémiologie Sociale, ERES, Sorbonne Université, Paris, France

* astrid.andersen@iplesp.upmc.fr

## Abstract

### Introduction

Results of the impact of lockdowns and stay-at-home orders during the COVID-19 pandemic on changes in cigarette smoking are mixed. Previous studies examining smoking changes during the early stages of the pandemic in 2020 have mainly focused on smoker's perception of changes in cigarette consumption. Such measure has not been widely used in other contexts, and therefore we aim to compare the discrepancy between smokers' perceived changes in cigarette smoking and the actual change in the number of cigarettes smoked, using repeated measurements.

### Methods

We included 134 smokers from the French TEMPO cohort with repeated measurements of their perceived changes in smoking habits during the first phase of the COVID-19 pandemic and the number of cigarettes smoked repeatedly from March to May 2020. We used generalized estimation equations (GEE) to examine the association between changes in the number of cigarettes smoked and the odds of mismatched answers.

### Results

The results suggest that at each study wave, 27–45% of participants provided mismatching answers between their perceived change in smoking habits and the actual change in the number of cigarettes smoked daily, measured repeatedly. Results from GEE analysis demonstrated that a mismatching assessment of smoking behavior was elevated among those who had an increase (OR = 2.52 [1.37;4.65]) or a decrease (OR = 5.73 [3.27;10.03]) in number of cigarettes smoked.

### Discussion

Our findings highlight the possibility of obtaining different results depending on how changes in tobacco smoking are measured. This highlights the risk of underestimating the actual

**Data Availability Statement:** Due to the personal questions asked in this study, research participants were guaranteed that all raw data will remain confidential. On reasonable request including

standards for General Data Protection Regulation data can be accessed, please send an email to cohort.tempo@inserm.fr. Anonymized data can only be shared after explicit approval of the French national committee for data protection for approval (Commission Nationale de l'Informatique et des Libertés, CNIL).

**Funding:** This research benefited from support from the French Institute for Public Health Research (IReSP) and the French Institute of Cancer (INCa), within the framework of the call for doctoral grants launched in 2020, under the project reference number "AAC-SPA-04". The TEMPO cohort is supported by the French National Research Agency (ANR) (ANR-Flash COVID), the French Insititue for Public Health Research IReSP (TGIR Cohortes), the French Institute of Cancer (INCa), the French Inter-departmental Mission for the Fight against Drugs and Drug Addiction (MILDeCA), and the Pfizer Foundation. The TEMPO COVID-19 study is supported by the European Union's Horizon 2020 research and innovation program RESPOND (funded under Horizon 2020 – the Framework Programme for Research and Innovation (2014-2020)). The content of this article reflects only the authors' views, and the European Community is not liable for any use that may be made of the information contained therein. The funders had no role in study design, data collection and analysis, decision to publish, or preparation of the manuscript.

**Competing interests:** The authors have declared that no competing interests exist.

changes in cigarette smoking during the COVID-19 pandemic, but also more generally when validating public health interventions or smoking cessation programs. Therefore, objective measures such as the actual consumption of psychoactive substances should be utilized, preferably on a longitudinal basis, to mitigate recall bias.

## Introduction

Since the onset of the COVID-19 pandemic in the spring of 2020, there has been a focus on changes in lifestyle habits due to pandemic-related rules and restrictions, including numerous international studies illustrating changes in smoking patterns. A recent systematic review and meta-analysis examining changes in tobacco smoking during the first pre-vaccination phases of the COVID-19 pandemic [1] concluded that the results are mixed. Literature from the early stages of the pandemic found no convergent results, neither in the number of persons who changed their behavior nor the direction of change [2–24], and later studies also reported this discrepancy [3, 25]. Indeed, varying percentages of people increasing their daily tobacco use have been found, ranging from 7% to 45% among smokers in general populations [2–11, 13–18, 20–24], as well as different values for the increase in the number of cigarettes smoked [2, 9]. In other contexts, no change or reduced tobacco consumption has been reported [19, 21].

In our view, the emerging literature has one major limitation. The way in which changes in smoking during the pandemic are measured in epidemiological studies varies and may contribute to an explanation of the different conclusions that have been drawn. In particular, since the COVID-19 pandemic led to major disruptions in regular life habits, it may have been particularly difficult for individuals to keep track of their smoking levels retrospectively. Most studies investigating changes in tobacco smoking are mainly based on self-reported survey questions about perceived changes in smoking habits [2–4, 6, 7, 9, 11, 13, 14, 16–18, 21–24, 26, 27], which is not a commonly used measure prior to the pandemic and could be imprecise [28]. In this study, we aim to examine the relationship between participants' perception of changes in smoking habits during the COVID-19 pandemic and the reported number of cigarettes smoked, measured longitudinally from March to May 2020.

## Material and methods

### Study population and procedures

Statistical analyses are based on data from participants in the French TEMPO cohort, a longitudinal study aiming to improve understanding of factors associated with mental health and psychoactive substance use, including tobacco smoking patterns [29]. The cohort was set-up in 2009 among young adults aged 22 to 35 years who had previously participated in a study on children's psychological problems in 1991 and 1999. Briefly, TEMPO participants were followed through self-completed questionnaires in 2009, 2011, 2015, and 2018 [29]. Between March and May 2020, TEMPO participants completed seven self-administered questionnaires (weekly for the first five questionnaires, and biweekly for questionnaire six and seven) aiming to better understand their experience of the COVID-19 epidemic, including changes in smoking levels. This aspect was named TEMPO COVID-19 study.

As the aim was to study changes in tobacco smoking, only smokers were included in this study.

The TEMPO cohort obtained approval from the French data protection authority (Commission Nationale de l'Informatique et des Libertés, CNIL, n˚ 908163). All participants provided their informed consent.

## Measures

**Tobacco smoking behavior.**   Throughout the seven data collection points, TEMPO COVID-19 participants were asked to record the number of cigarettes smoked daily or weekly. Additionally, they were asked whether they had changed their smoking behavior within the preceding seven days ("In the last 7 days, has your consumption of tobacco products changed?" with response options including 'No', 'Yes, it decreased', 'Yes, I stopped smoking', 'Yes, I relapsed to smoking after quitting', 'Yes, it increased', 'Yes, I started smoking'). In this paper, we will use the term *perceived change* to denote this measure.

**Outcome: Match/mismatch variable.**   To examine the extent of discrepancy between participants' assessment of changes in smoking levels and the actual change in the reported number of cigarettes smoked, our outcome variable captures a match vs. mismatch between perceived changes in smoking patterns and the actual change in the number of cigarettes participants reported smoking between two waves of data collection. Therefore, we calculated the percentage change in smoking by subtracting the number of cigarettes reported in one particular week from the number reported in the preceding week. To the best of our knowledge, there is no consensus in the scientific literature on a meaningful threshold to define changes in tobacco consumption. Given that the average number of cigarettes smoked per day was 7.0 (standard deviation = 6.8) in our sample, we opted for a 25% change (equivalent to 2 cigarettes). Participants were considered to have altered their smoking levels if the number of cigarettes smoked varied by at least 25%, resulting in one of the following classifications: 1) no change, 2) increase, or 3) decrease. In additional analyses, we used other definitions of changes in smoking patterns ($\geq$10% and $\geq$50% of cigarettes smoked). If no information was available in the preceding week, the last available data were used. Hereafter, we compared whether the percentage change in the number of cigarettes smoked was consistent with participants' perceived changes in smoking. If the two variables were consistent, the outcome was a match; if not, we considered it to be a mismatch.

**Covariates.**   To study factors associated with a mismatch in smoking assessments, our statistical analyses controlled for factors potentially associated with tobacco smoking, including sex, age, marital status (single, divorced or widowed vs. married, civil union or in a relationship), education level (Bac+2 or lower vs. Bac+3 or higher), employment status (unemployed vs. employed), type of employment (unstable vs. stable), household income ($\leq$2500 euros vs. $\geq$2501 euros), and symptoms of depression (yes vs. no), measured with items from the depressive syndrome scale based on the Adult Self Report (ASR)-Achenbach System [30].

## Statistical analysis

First, we described the distribution of matching and mismatching answers by study wave. Second, to examine the longitudinal association between mismatching answers and the actual change in the number of cigarettes smoked, we used Generalized Estimation Equations (GEE) with a logit link and an exchangeable model fit for both bivariate and multivariate models. In the final multivariate model, we included variables associated with the outcome with a p-value < 0.20 in bivariate analyses (S1 Table). To account for differential attrition, we estimated inverse probability weights (IPW) by estimating non-response among study participants included in our statistical analyses compared to the original TEMPO cohort, and weighted bivariate and multivariate GEE regression models. In sensitivity analyses, we used a 10% and

50% change in the number of cigarettes smoked to define the study outcome. All statistical analyses were performed using SAS® (version 9.4).

## Results

Our study included 134 persons who reported daily or occasional tobacco smoking and completed at least one TEMPO COVID-19 questionnaire. Participants were predominantly female (60%), were on average 40 years old, married or in a relationship (78%), and the majority had a household income exceeding 2500 euros (73%) (S1 Table).

As shown in Table 1, in all seven study waves, 21–30% of participants perceived an increase in their smoking behavior, while 8–25% reported a decrease, and more than 50% reported no change. When examining the actual change in the number of cigarettes smoked, 17–31% of participants increased their consumption, 13–27% decreased, and 50–60% reported no change. However, at each study wave, 27–45% provided mismatching answers. Among participants with mismatching assessments of smoking, the majority did not perceive a change, despite actually reporting a change in the number of cigarettes smoked.

Results from the multivariate GEE model are presented in Table 2, and demonstrate that a mismatching assessment of smoking behavior reflected more of a decrease in cigarette consumption (OR = 5.73, 95% confidence interval (CI) = [3.27;10.03]) than an increase (OR = 2.52, 95%CI = [1.37;4.65]). Supplementary analyses yielded similar results using a 10% cutoff for change in smoking behavior. When using a 50% cutoff for change in smoking behavior, marital status was additionally associated with a mismatching assessment; compared to individuals married or in a partnership, those who were single, separated, or divorced had an odds ratio of 2.09 (95% CI = [1.07;4.08]).

## Discussion

This study highlights the possibility of mismatching assessments of smoking behavior depending on the type of measure used. This situation often reflects a perception of no change, while the reported number of cigarettes smoked differs from previous assessments. At the same time, people who reduce their consumption are more likely to report mismatching answers than those with unchanged consumption, suggesting that a decrease in the number of cigarettes smoked is not perceived. Persons not in stable relationships, and therefore likely living alone, had an elevated probability of reporting different perceptions and actual changes in smoking levels. Misperceptions of smoking patterns may have been exacerbated in the context of the COVID-19 pandemic, as individuals' daily routines and habits were modified, possibly making it difficult to accurately assess their own behavior. This should be considered when studying self-reported changes in smoking behavior.

The interpretation of our results needs to be considered in light of several possible limitations. First, we studied a relatively small sample of smokers, most of whom had a high socioeconomic position, which may limit the generalizability of our results [29]. Nevertheless, there is no previous evidence showing that the difference between perceived and actual change in smoking patterns may vary depending on these characteristics. However, it may be that in a larger study, other factors associated with a mismatch in smoking assessment may have become apparent. Second, for respondents who did not answer all seven waves of data collection, we used the last available measurement of the number of cigarettes, which may have induced measurement error. Nevertheless, the time between each study wave was very limited (7–15 days), and this should not be a major source of bias. Third, the smoking measures we used are self-reported and may be subject to misclassification. However, self-reported smoking status is considered accurate when compared with biomarkers and biochemical measures of

**Table 1. Descriptive distribution of smoking behavior and match/mismatch per wave including 1) responses to perceived changes in smoking behavior, 2) actual changes in the number of cigarettes smoked (+/- 25%), 3) the number of individuals with matching responses, 4) perceived changes among those with matching responses, 5) the number of individuals with mismatched responses, and 6) individuals' perceived changes among those with mismatched responses.** TEMPO cohort study, March to May 2020 (n = 134).

| | | 1 | 2 | 3 | 4 | 5 | 6 |
|---|---|---|---|---|---|---|---|
| | | Perceived changes n (%) | Actual changes n (%) | Match n (%) | Perceived responses among matches n (%) | Mismatch n (%) | Perceived responses among mismatches n (%) |
| Wave 1 | No change | 25 (50%) | 27 (53%) | 30 (59%) | 18 (60%) [a] | 21 (41%) | 7 (33%) [d] |
| n = 51 | Decreased | 13 (25%) | 14 (27%) | | 7 (23%) [b] | | 6 (29%) [e] |
| | Increased | 13 (25%) | 10 (20%) | | 5 (17%) [c] | | 8 (38%) [f] |
| Wave 2 | No change | 50 (57%) | 41 (47%) | 51 (58%) | 29 (57%) [a] | 37 (42%) | 21 (57%) [d] |
| n = 88 | Decreased | 13 (15%) | 22 (25%) | | 7 (14%) [b] | | 6 (16%) [e] |
| | Increased | 25 (28%) | 25 (28%) | | 15 (29%) [c] | | 10 (27%) [f] |
| Wave 3 | No change | 50 (60%) | 41 (49%) | 51 (61%) | 31 (61%) [a] | 32 (39%) | 19 (59%) [d] |
| n = 83 | Decreased | 8 (10%) | 16 (19%) | | 5 (10%) [b] | | 3 (9%) [e] |
| | Increased | 25 (30%) | 26 (31%) | | 15 (29%) [c] | | 10 (31%) [f] |
| Wave 4 | No change | 51 (63%) | 54 (67%) | 59 (73%) | 43 (73%) [a] | 22 (27%) | 8 (36%) [d] |
| n = 81 | Decreased | 10 (12%) | 13 (16%) | | 6 (10%) [b] | | 4 (18%) [e] |
| | Increased | 20 (25%) | 14 (17%) | | 10 (17%) [c] | | 10 (45%) [f] |
| Wave 5 | No change | 54 (70%) | 45 (58%) | 47 (61%) | 36 (77%) [b] | 30 (39%) | 18 (60%) [d] |
| n = 77 | Decreased | 7 (9%) | 16 (21%) | | 4 (8%) [b] | | 3 (10%) [e] |
| | Increased | 16 (21%) | 16 (21%) | | 7 (15%) [c] | | 9 (30%) [f] |
| Wave 6 | No change | 46 (69%) | 41 (61%) | 37 (55%) | 31 (84%) [b] | 30 (45%) | 15 (50%) [d] |
| n = 67 | Decreased | 7 (10%) | 12 (18%) | | 2 (5%) [b] | | 5 (17%) [e] |
| | Increased | 14 (21%) | 14 (21%) | | 4 (11%) [c] | | 10 (33%) [f] |
| Wave 7 | No change | 40 (67%) | 39 (65%) | 39 (65%) | 30 (77%) [a] | 21 (35%) | 10 (48%) [d] |
| n = 60 | Decreased | 5 (8%) | 8 (13%) | | 2 (5%) [b] | | 3 (14%) [e] |
| | Increased | 15 (25%) | 13 (22%) | | 7 (18%) [c] | | 8 (38%) [f] |

[a] Perception of no change in tobacco consumption and no reported change in the number of cigarettes

[b] Perception of a decrease in tobacco consumption matching a decrease in the number of cigarettes

[c] Perception of an increase in tobacco consumption matching an increase in the number of cigarettes

[d] Perception of no change in tobacco consumption but either a decrease or increase in the number of cigarettes

[e] Perception of a decrease in tobacco consumption but either no change or an increase in the number of cigarettes

[f] Perception of an increase in tobacco consumption but either no change or a decrease in the number of cigarettes

tobacco use [31–33]. We chose a cutoff of 25% based on the relatively low mean of cigarettes smoked daily to ensure that a change in the number of cigarettes smoked would minimize the risk of misclassification. Sensitivity analyses showed similar results. Our study also has several strengths that outweigh the previously mentioned limitations. First, we used longitudinal data to compare perceived changes in tobacco use and the number of cigarettes smoked across seven waves of measurements, thus limiting recall bias. Second, both perceived and actual changes in smoking behavior were measured simultaneously.

**Table 2. Multivariate GEE models showing the odds of a mismatch between perceived and actual changes in smoking: TEMPO cohort (n = 134).**

| | | 25% change[1] | | 10% change[1] | | 50% change[1] | |
|---|---|---|---|---|---|---|---|
| | | OR | p | OR | p | OR | p |
| **Change in the number of cigarettes** | | | | | | | |
| | Yes, increased (≥25%) | **2.52 [1.37;4.65]** | 0.0030 | **1.94 [1.15;3.27]** | 0.0126 | **2.43 [1.43;4.13]** | 0.0010 |
| | Yes, decreased (≥25%) | **5.73 [3.27;10.03]** | <0.0001 | **4.31 [2.61;7.12]** | <0.0001 | **2.96 [1.66;5.28]** | 0.0002 |
| | No | Ref | | Ref | | Ref | |
| **Marital status** | | | | | | | |
| | Single, divorced, or widowed | 1.30 [0.82;2.08] | 0.2668 | 0.98 [0.65;1.49] | 0.9416 | **2.09 [1.07;4.08]** | 0.0303 |
| | Married or in a couple | Ref | | Ref | | Ref | |
| **Type of employment** | | | | | | | |
| | Unstable | 1.29 [0.72;2.34] | 0.3939 | 1.11 [0.65;1.89] | 0.6953 | 1.24 [0.59;2.58] | 0.5714 |
| | Stable | Ref | | Ref | | Ref | |
| **Symptoms of depression** | | | | | | | |
| | Yes | 1.63 [0.50;1.14] | 0.0699 | 1.14 [0.67;1.94] | 0.6193 | 1.43 [0.82;2.48] | 0.2048 |
| | No | Ref | | Ref | | Ref | |

[1]The three GEE models have different percentage cutoffs in the actual change in the number of cigarettes to calculate a mismatch

Our study reveals that more than one-third of study participants provided inconsistent answers when asked about perceived changes in tobacco use during the COVID-19 pandemic and the number of cigarettes smoked. While changes in tobacco consumption in the context of the COVID-19 pandemic have been previously reported, the direction of change and associated factors vary considerably. Through this study, we demonstrate that the method of measuring changes in tobacco consumption may impact our findings.

As previous mentioned, there has been an increased focus on studying changes in tobacco consumption during the COVID-19 pandemic, with most studies relying on perceptions of change phrased as 'Compared to pre-lockdown, has your smoking consumption changed?' with response categories like 'It has increased/it has remained the same/it has decreased' [2, 5, 14, 17, 21, 24]. As observed in our study, participants' subjectivity might explain some of the differences across studies conducted during the COVID-19 pandemic. Furthermore, the use of retrospectively collected data may increase the risk of recall bias [34] and result in biased estimates and associations. The reported number of cigarettes may be highly influenced by the perception of change and the time since the beginning of lockdown. When studying changes in working time before and during the pandemic, the use of retrospective questions has been found to lead to an underestimation of change between the current and pre-pandemic situation [28]. This trend is likely to be present in our study as well, where a substantial group of participants perceives no change, while longitudinal measures indicate their behavior did change. Therefore, the results of this paper must be seen in the light of the assumption that repeated measures of the number of cigarettes smoked are more valid than perceived change, particularly when reported retrospectively [28].

## Conclusion

To our knowledge, ours is the first study to examine the match and mismatch between perceived and actual changes in tobacco consumption during the COVID-19 pandemic. Our findings highlight the possibility of obtaining different results depending on to the way changes in tobacco smoking are measured. This highlights the risk of underestimating the actual change in cigarette smoking during the first two months of the COVID-19 pandemic.

The results of this study should be considered when planning future data collection aimed at studying changes in cigarette smoking. Questions related to perceived change and cross-sectional measurement, including both a pre- and post-pandemic dimension, increase the risk of underestimating actual changes in smoking patterns, which may lead to biased results and conclusions. Examining changes in smoking may be more complex and cannot be based only on smokers' own perception of change. Instead, objective measures such as the actual consumption of psychoactive substances before and after should be used, preferably on a longitudinal basis, to mitigate recall bias, especially when validating public health interventions or smoking cessation programs.

## Supporting information

**S1 Table. Socio-demographic characteristics of study population and unadjusted associations with a mismatching tobacco use assessment.** TEMPO cohort (n = 134), GEE model (OR, 95% CI).
(DOCX)

## Acknowledgments

We would like to thank all the TEMPO participants who provided data for this project.

## Author Contributions

**Conceptualization:** Astrid Juhl Andersen, Maria Melchior, Murielle Mary-Krause.

**Data curation:** Solène Wallez.

**Formal analysis:** Astrid Juhl Andersen.

**Investigation:** Astrid Juhl Andersen, Murielle Mary-Krause.

**Methodology:** Murielle Mary-Krause.

**Project administration:** Astrid Juhl Andersen.

**Supervision:** Maria Melchior, Murielle Mary-Krause.

**Validation:** Murielle Mary-Krause.

**Writing – original draft:** Astrid Juhl Andersen.

**Writing – review & editing:** Solène Wallez, Maria Melchior, Murielle Mary-Krause.

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
