## [Decision Letter · Decision Letter 0]

4 Jan 2024

PONE-D-23-36304Our perception may not be reality: A longitudinal study of the relationship between perceived and actual change in smoking behaviorPLOS ONE

Dear Dr. Andersen,

Thank you for submitting your manuscript to PLOS ONE. After careful consideration, we feel that it has merit but does not fully meet PLOS ONE’s publication criteria as it currently stands. Therefore, we invite you to submit a revised version of the manuscript that addresses the points raised during the review process.

We look forward to receiving your revised manuscript.

Kind regards,

Suzit Bhusal

Academic Editor

PLOS ONE

Journal Requirements:

3. In the online submission form, you indicated that [Due to the personal questions asked in this study, research participants were guaranteed that all raw data will remain confidential. On reasonable request including standards for General Data Protection Regulation data can be accessed, please send an email to cohort.tempo@inserm.fr. Anonymized data can only be shared after explicit approval of the French national committee for data protection for approval (Commission Nationale de l’Informatique et des Libertés, CNIL).]. 

4. Please include your tables as part of your main manuscript and remove the individual files. Please note that supplementary tables (should remain/ be uploaded) as separate "supporting information" files.

Reviewers' comments:

Reviewer's Responses to Questions

**Comments to the Author**

1. Is the manuscript technically sound, and do the data support the conclusions?

Reviewer #1: Yes

Reviewer #2: Yes

2. Has the statistical analysis been performed appropriately and rigorously? 

Reviewer #1: Yes

Reviewer #2: I Don't Know

3. Have the authors made all data underlying the findings in their manuscript fully available?

Reviewer #1: No

Reviewer #2: No

4. Is the manuscript presented in an intelligible fashion and written in standard English?

Reviewer #1: Yes

Reviewer #2: Yes

5. Review Comments to the Author

Reviewer #1: Abstract:

The abstract provides a clear overview of the study, summarizing the research question, methodology, and key findings. It effectively communicates the main objectives and results of the research. However, it could benefit from a brief mention of the practical implications of the findings.

Introduction:

1. The introduction sets the stage well by highlighting the importance of studying changes in lifestyle habits, particularly smoking, during the COVID-19 pandemic. It appropriately references existing literature, providing a comprehensive background.

2. The introduction effectively states the research gap regarding the varied results in previous studies and introduces the primary objective of comparing perceived and actual changes in smoking behavior.

3. Consider clarifying the term "pre-vaccination period" for better contextualization, as the study seems to focus on the early stages of the pandemic.

Methods:

1. The methods section is comprehensive, detailing the study population, procedures, measures, and statistical analyses.

2. Clarify the rationale behind choosing the specific cut-off percentages (25%, 10%, and 50%) for defining changes in smoking behavior. Provide a brief justification for these choices.

3. The inclusion of sensitivity analyses enhances the robustness of the study.

Discussion:

1. The discussion effectively interprets the findings, emphasizing the potential impact of measurement methods on the perception of changes in smoking behavior.

2. The study's limitations are appropriately acknowledged, adding transparency and credibility to the research.

3. Consider expanding on the practical implications of the findings for public health interventions or smoking cessation programs.

General Comments:

1. The manuscript is well-written, and the language is clear and concise.

2. The article could benefit from a brief statement in the abstract or conclusion about the broader implications of the findings for public health or smoking cessation efforts.

3. Ensure consistency in terminology and abbreviations throughout the manuscript.

Reviewer #2: General overview

The study investigates the discrepancy between smokers' perceived changes in smoking habits and their actual change in the number of cigarettes smoked during the COVID-19 pandemic. Using longitudinal data from the French TEMPO cohort, the authors explore how smokers' perceptions align with their reported smoking behavior.

Strengths:

addresses an important gap by examining the variance between perceived and actual changes in smoking behavior during the pandemic, offering insights into measurement discrepancies.

Longitudinal Approach

The use of Generalized Estimation Equations (GEE) offers a robust statistical method to examine the association between perceived and actual changes in smoking habits.

Proper discussions of limitations and strength

Revisions

I was not able to find the tables mentioned in the manuscript. Please ensure the tables are provided in the revision so I can properly review the manuscript.

The authors could elaborate on the implications of the findings in future research and emphasize the significance of objective measures in evaluating smoking behavior further.

In the Measures section in Material and Methods

You mention that The cutoffs to define the change in smoking patterns were chosen based on the average number of daily cigarettes among the study population.

Please clarify this statement.

There were a few grammatical errors in the manuscript. Please conduct a thorough proofreading of the manuscript to rectify any errors. For example:

1. In the Second Paragraph of the Results section

participants perceived an increased in their smoking behavior

To

Participants perceived an increase in their smoking behavior

2. In the second paragraph of the conclusion section

actual change in smoking patterns may vary depending these characteristics.

To

actual change in smoking patterns may vary depending on these characteristics.

3. In the discussion

As observed in our study, participants’ subjectivity might explain some of the differences across studied conducted during the COVID-19 pandemic.

To

As observed in our study, participants’ subjectivity might explain some of the differences across studies conducted during the COVID-19 pandemic.

6. PLOS authors have the option to publish the peer review history of their article (what does this mean?). If published, this will include your full peer review and any attached files.

Reviewer #1: **Yes: **Ashlesha Chaudhary

Reviewer #2: No

---

## [Author Response · Author response to Decision Letter 0]

22 Feb 2024

Reviewer #1

Abstract:

The abstract provides a clear overview of the study, summarizing the research question, methodology, and key findings. It effectively communicates the main objectives and results of the research. However, it could benefit from a brief mention of the practical implications of the findings.

 We thank the reviewer for her positive comments. To address the reviewer’s request, we revised the abstract to incorporate the practical implications of our findings as follows: «This highlights the risk of underestimating the actual changes in cigarette smoking during the COVID-19 pandemic, but also more generally when validating public health interventions or smoking cessation programs. Therefore, objective measures such as the actual consumption of psychoactive substances should be utilized, preferably on a longitudinal basis, to mitigate recall bias» (page 3, lines 52-54).

Introduction:

1. The introduction sets the stage well by highlighting the importance of studying changes in lifestyle habits, particularly smoking, during the COVID-19 pandemic. It appropriately references existing literature, providing a comprehensive background.

 We thank the reviewer for her positive comment.

2. The introduction effectively states the research gap regarding the varied results in previous studies and introduces the primary objective of comparing perceived and actual changes in smoking behavior.

 We thank the reviewer for her positive comment.

3. Consider clarifying the term "pre-vaccination period" for better contextualization, as the study seems to focus on the early stages of the pandemic.

 We agree with the reviewer that the term «pre-vaccination period» is not clear. We have modified it with «the early stages of the pandemic» in the introduction of the abstract (page 2, line 30).

Methods:

1. The methods section is comprehensive, detailing the study population, procedures, measures, and statistical analyses.

 We thank the reviewer for her positive comment.

2. Clarify the rationale behind choosing the specific cut-off percentages (25%, 10%, and 50%) for defining changes in smoking behavior. Provide a brief justification for these choices.

 There is no consensus in the literature on how to define a meaningful threshold in change in tobacco consumption. Since the average number of cigarettes smoked per day was 7 in our sample, a threshold of 25% (equivalent to 2 cigarettes) appeared reasonable to consider. However, we also tested other threshold values to ensure result consistency. We revised the Measures/Outcome section to provide a clearer explanation of this rationale, as follows: «To the best of our knowledge, there is no consensus in the scientific literature on a meaningful threshold to define changes in tobacco consumption. Given that the average number of cigarettes smoked per day was 7.0 (standard deviation=6.8) in our sample, we opted for a 25% change (equivalent to 2 cigarettes). Participants were considered to have altered their smoking levels if the number of cigarettes they smoked varied by at least 25%, resulting in one of the following classifications: 1) no change, 2) increase, or 3) decrease», (pages 6-7, lines 125-132).

3. The inclusion of sensitivity analyses enhances the robustness of the study.

 We thank the reviewer for her positive comment.

Discussion:

1. The discussion effectively interprets the findings, emphasizing the potential impact of measurement methods on the perception of changes in smoking behavior.

 We thank the reviewer for her positive comment.

2. The study's limitations are appropriately acknowledged, adding transparency and credibility to the research.

 We thank the reviewer for her positive comment.

3. Consider expanding on the practical implications of the findings for public health interventions or smoking cessation programs.

 According to the reviewer’s comment, we revised the conclusion of our study to incorporate the practical implications of the findings for public health as follows: «Instead, objective measures such as the actual consumption of psychoactive substances before and after should be used, preferably on a longitudinal basis, to mitigate recall bias, especially when validating public health interventions or smoking cessation programs» (page 13, lines 260-263).

General Comments:

1. The manuscript is well-written, and the language is clear and concise.

 We thank the reviewer for her positive comment.

2. The article could benefit from a brief statement in the abstract or conclusion about the broader implications of the findings for public health or smoking cessation efforts.

 According to the reviewer’s comment, we revised the abstract and the conclusion to incorporate the practical implications of the findings for public health (page 3, lines 50-54 and page 13, lines 260-263)

3. Ensure consistency in terminology and abbreviations throughout the manuscript.

 Following the reviewer's comment, the article was corrected by a native English speaker.

 

Reviewer #2

General overview

The study investigates the discrepancy between smokers' perceived changes in smoking habits and their actual change in the number of cigarettes smoked during the COVID-19 pandemic. Using longitudinal data from the French TEMPO cohort, the authors explore how smokers' perceptions align with their reported smoking behavior.

Strengths: addresses an important gap by examining the variance between perceived and actual changes in smoking behavior during the pandemic, offering insights into measurement discrepancies.

Longitudinal Approach: The use of Generalized Estimation Equations (GEE) offers a robust statistical method to examine the association between perceived and actual changes in smoking habits.

Proper discussions of limitations and strength

 We thank the reviewer for his.her positive comment.

Revisions

I was not able to find the tables mentioned in the manuscript. Please ensure the tables are provided in the revision so I can properly review the manuscript.

 We apologize for the inconvenience caused by the reviewer's inability to locate the tables in the article. They were indeed included in the supplementary data. We have now included our tables as part of our main manuscript, immediately after the first paragraph in which they are cited (pages 9-10), and the supplementary data are provided as a separate file, in accordance with the recommendation from Plos One.

The authors could elaborate on the implications of the findings in future research and emphasize the significance of objective measures in evaluating smoking behavior further.

 According to the reviewer’s comment, we revised the conclusion to incorporate the practical implications of the findings for public health as follows: «Instead, objective measures such as the actual consumption of psychoactive substances before and after should be used, preferably on a longitudinal basis, to mitigate recall bias, especially when validating public health interventions or smoking cessation programs» (page 13, lines 260-263). We also revised the abstract to incorporate the practical implications of the findings as follows: «This highlights the risk of underestimating the actual changes in cigarette smoking during the COVID-19 pandemic, but also more generally when validating public health interventions or smoking cessation programs. Therefore, objective measures such as the actual consumption of psychoactive substances should be utilized, preferably on a longitudinal basis, to mitigate recall bias» (page 3, lines 50-54).

In the Measures section in Material and Methods: You mention that The cutoffs to define the change in smoking patterns were chosen based on the average number of daily cigarettes among the study population. Please clarify this statement.

 There is no consensus in the scientific literature on a meaningful threshold to define changes in tobacco consumption. Since the average number of cigarettes smoked per day was 7 in our sample, a threshold of 25% (equivalent to 2 cigarettes) appeared reasonable to consider. However, we also tested other threshold values to ensure result consistency. We revised the Measures/Outcome section to provide a clearer explanation of this rationale, as follows: «To the best of our knowledge, there is no consensus in the scientific literature on a meaningful threshold to define changes in tobacco consumption. Given that the average number of cigarettes smoked per day was 7.0 (standard deviation=6.8) in our sample, we opted for a 25% change (equivalent to 2 cigarettes). Participants were considered to have altered their smoking levels if the number of cigarettes smoked varied by at least 25%, resulting in one of the following classifications: 1) no change, 2) increase, or 3) decrease», (pages 6-7, lines 125-132).

There were a few grammatical errors in the manuscript. Please conduct a thorough proofreading of the manuscript to rectify any errors. For example:

1. In the Second Paragraph of the Results section

participants perceived an increased in their smoking behavior

To

Participants perceived an increase in their smoking behavior

2. In the second paragraph of the discussion section

actual change in smoking patterns may vary depending these characteristics.

To

actual change in smoking patterns may vary depending on these characteristics.

3. In the discussion

As observed in our study, participants’ subjectivity might explain some of the differences across studied conducted during the COVID-19 pandemic.

To

As observed in our study, participants’ subjectivity might explain some of the differences across studies conducted during the COVID-19 pandemic.

 Following the reviewer's comment, the article was corrected by a native English speaker.

---

## [Editor Report · Decision Letter 1]

18 Mar 2024

Our perception may not be reality: A longitudinal study of the relationship between perceived and actual change in smoking behavior

PONE-D-23-36304R1

Dear Dr. Andersen,

We’re pleased to inform you that your manuscript has been judged scientifically suitable for publication and will be formally accepted for publication once it meets all outstanding technical requirements.

Within one week, you’ll receive an email detailing the required amendments. When these have been addressed, you’ll receive a formal acceptance letter and your manuscript will be scheduled for publication.

Kind regards,

Academic Editor

PLOS ONE

---

## [Editor Report · Acceptance letter]

21 Mar 2024

PONE-D-23-36304R1 

PLOS ONE

Dear Dr. Andersen, 

I'm pleased to inform you that your manuscript has been deemed suitable for publication in PLOS ONE. Congratulations! Your manuscript is now being handed over to our production team.

Kind regards, 

on behalf of

Dr. Suzit Bhusal 

Academic Editor

PLOS ONE